# Multi-Scale and Multi-Match for Few-Shot Plant Disease Image Semantic Segmentation

**Wenji Yang [1], Wenchao Hu [1], Liping Xie [2] and Zhenji Yang [3],***

1. School of Software, Jiangxi Agricultural University, Nanchang 330045, China
2. School of Computer and Information Engineering, Jiangxi Agricultural University, Nanchang 330045, China
3. Finance Office, Jiangxi Agricultural University, Nanchang 330045, China
*  Correspondence: yzhenji1115@163.com

**Abstract:** Currently, deep convolutional neural networks have achieved great achievements in semantic segmentation tasks, but existing methods all require a large number of annotated images for training and do not have good scalability for new objects. Therefore, few-shot semantic segmentation methods that can identify new objects with only one or a few annotated images are gradually gaining attention. However, the current few-shot segmentation methods cannot segment plant diseases well. Based on this situation, a few-shot plant disease semantic segmentation model with multi-scale and multi-prototypes match (MPM) is proposed. This method generates multiple prototypes and multiple query feature maps, and then the relationships between prototypes and query feature maps are established. Specifically, the support feature and query feature are first extracted from the high-scale layers of the feature extraction network; subsequently, masked average pooling is used for the support feature to generate prototypes for a similarity match with the query feature. At the same time, we also fuse low-scale features and high-scale features to generate another support feature and query feature that mix detailed features, and then a new prototype is generated through masked average pooling to establish a relationship with the query feature of this scale. Subsequently, in order to solve the shortcoming of traditional cosine similarity and lack of spatial distance awareness, a CES (cosine euclidean similarity) module is designed to establish the relationship between prototypes and query feature maps. To verify the superiority of our method, experiments are conducted on our constructed PDID-5$^i$ dataset, and the mIoU is 40.5%, which is 1.7% higher than that of the original network.

**Keywords:** few-shot semantic segmentation; mixed similarity; multi-scale fusion; plant disease





## 1. Introduction

With the rapid development of agricultural information technology [1–6], semantic segmentation using FCN [7], UNet [8], SegNet [9], Deeplab [10], ASPP [11] has become one of the main technologies of agricultural intelligence. However, large pixel-by-pixel annotated datasets, which are costly to be obtained, are required to train these models. Although weakly supervised learning can reduce this cost to some extent, it still requires a lot of weakly annotated data. To solve the problem of obtaining a large number of annotated datasets, few-shot semantic segmentation [12], which aims to learn from a few support images, is proposed and has gradually attracted attention in various fields, especially in plant disease segmentation, making it discriminative for new unseen classes.

Most few-shot segmentation methods learn from a small number of support images and then feed the learned knowledge into a parameterized module for query segmentation. However, this approach resulted in a mixture of model segmentation and supported semantic features due to the simultaneous feature extraction and object segmentation process. At the same time, the current few-shot semantic segmentation method is generally implemented by comparing the similarity between prototypes and query features. For example, SG-one [13] extracts the guiding feature of the support image through VGG16 [14]

and then uses the cosine similarity to establish the relationship between the prototypes and the query image feature. Taking this research idea, the PANet [15] adds support prototypes and query features, which are regularized to provide better generalization ability.

Although few-shot semantic segmentation has made great progress in processing natural images, these methods cannot handle the segmentation of plant disease images well. Because the guidance feature extracted from the support images through the backbone network and the query image is used for foreground detection, it cannot handle the huge differences in the shape and texture of different plant diseases well. Therefore, simply matching the prototypes generated from the deep feature of the supported image with the query image will lose many disease features, which can lead to predictions that ignore smaller disease areas and fail to identify differences among multiple diseases accurately.

To overcome the above problem of few-shot segmentation algorithms in plant disease images, a few-shot semantic segmentation network based on multi-scale and multi-match plant disease images is proposed. In this paper, semantic segmentation is performed for early disease images of plant leaves, and the diseased areas in leaves are drawn, which provides a new method for plant disease control. Specifically, the high-scale support feature and query feature are obtained through the last layer of the feature extraction network, and then the feature relationship is established to obtain the similarity map. In order to obtain more detailed features, this paper fuses the low-scale and high-scale features extracted from the feature extraction network VGG16 to obtain fused support features and fused query features that contain the more detailed features. Then, new prototypes generated from the fused support feature by masked average pooling are used to match with the fused query feature for similarity. Finally, the similarity map is obtained through multi-scale fusion. Specifically, the average similarity is obtained by averaging multiple similarities. By matching multi-support image prototypes and multi-query image features, the recognition accuracy of the network model can be improved to a certain extent.

Now, the cosine similarity is used in the original algorithm for matching the prototypes of the supported image with the query feature, which calculates the similarity between two vectors only by considering the similarity of their direction angles in the space and does not consider the distance between the two vectors. Therefore, a hybrid similarity calculation is adopted, which calculates the euclidean distance and the cosine similarity of the two vectors. Then a weighted sum is performed according to 9:1 (the cosine similarity: the euclidean distance). In this way, the method can obtain more accurate similarity maps.

With limited computing power, it is important to allocate computing resources to more important tasks. In deep learning, with the increase of network parameters, there will be a problem of information overload. Therefore, the CBAM (convolutional block attention module) [16] is introduced into our network after the fusion of shallow features and deep features, which can not only pay more attention to important information and filter other irrelevant information but also improve the efficiency and performance of segmentation tasks.

There are few plant disease datasets suitable for few-shot semantic segmentation tasks. Therefore, this paper constructs a plant disease dataset (PDID-5$^i$) containing ten different categories, which are then annotated at the pixel level, and conducts experiments on the dataset to verify the effectiveness of our network.

The main contributions of our method are as follows:

1.  Multi-scale and multi-prototypes match is proposed for few-shot plant disease semantic segmentation. This method generates multiple prototypes and multiple query feature maps at different scales, and then the relationships between prototypes and query feature maps are established through the similarity measure method. Finally, the relationships at different scales are fused. With this approach, our network can more precisely identify plant disease signatures.
2.  The mixed similarity is designed as the weighted sum of cosine similarity and euclidean distance. When the similarity of the direction and the actual distance between two vectors are jointly considered, more accurate similarity can be obtained.

3.  A CBAM attention module is added to our network to make the network pay attention to the important plant disease feature and ignore interference information, which is beneficial to improve accuracy.
4.  To accomplish the few-shot semantic segmentation task, we constructed a plant disease dataset(PDID-5$^i$) that is suitable for the task. Experiments on the dataset show that the model we designed is very effective.

## 2. Related Work

### 2.1. Semantic Segmentation of Plant Disease Images

Semantic segmentation is an extremely challenging task whose purpose is to perform pixel-wise class prediction. In recent years, with the rapid development of deep learning, semantic segmentation of plant disease images has gradually received attention [17–20]. Some works utilize a combination of traditional methods and convolutional neural networks to solve the disease segmentation task [21,22]. For example, Sodjinou et al. [23] proposed a segmentation method based on the combination of semantic segmentation and the K-means algorithm. Through feature fusion structure, the most differential information can be obtained from multiple original feature sets involved in fusion, thus improving the accuracy of model recognition. As Lin et al. [24] used U-net convolutional neural network to fuse the feature from the encoding stage and the feature from the decoding stage and then segmented the cucumber powdery mildew. Zhong et al. [25] proposed a three-stream segmentation network. First, the hole convolution was used to expand the receptive field of each branch, and the feature fusion module was used to fuse the feature of each branch to obtain rich context information, thereby improving the segmentation accuracy. He et al. [26] proposed a lightweight network based on multi-scale feature fusion and attention refinement to enhance the representation ability of deep networks to extract feature maps. Along this direction, we fuse the detail feature in the shallow layers and the semantic feature in the deep layers to generate the fused feature maps and employ the attention mechanism to refine the feature maps. However, few-shot semantic segmentation can solve the problem of difficulty in obtaining agricultural datasets based on traditional semantic segmentation.

### 2.2. Few-Shot Semantic Segmentation

The few-shot semantic segmentation task uses a small number of annotated query images with a new category. Zhang et al. [13] proposed an efficient similarity-guided network to obtain the guided feature of the support image and then used the cosine similarity to match the query image feature to complete the few-shot segmentation task. Tang et al. [27] designed a CRE contextual relation encoder to capture the local relation between foreground and background and repeatedly used the CRE module to optimize the segmentation module. Wang et al. [28] introduced a democratized graph attention mechanism, which can activate more pixels on objects when the relationship between the support image and the query image is established. Therefore, the method can transfer more guiding features from the supported image to the query image, improving the model performance and robustness. Xie et al. [29] proposed a scale-aware graph neural network to construct a scale-aware graph using multi-scale query feature maps, in which support image guidance is employed as nodes of the graph. However, this method has poor flexibility and scalability and is not necessarily suitable for plant diseases. Few-shot segmentation can be used in plant disease segmentation to solve the problem that plant-labeled images are difficult to obtain from reality.

## 3. The Proposed Method

### 3.1. Problem Setting

The few-shot segmentation model proposed in this paper aims to obtain the guiding feature from a small number of annotated support images to segment the new segmented objects of the query image. This paper adopts the following strategies to train and test the

model. First, we divide the dataset category set into the known category set $C_{know}$ and the unknown category set $C_{unknow}$. The data set $D_{train}$ for training is from $C_{know}$, and the test set $D_{test}$ is constructed from $C_{unknow}$. Finally, we train the model on the training set and evaluate the model performance on the test set with the trained model.

The training set and the test set consist of several episodes, each containing an annotated support image set $S_i$ and an unannotated query image set $Q_i$. In the k-shot semantic segmentation task setting, each semantic category in the support set $S_i$ has K pairs of <image, mask>. At the same time, the $C_{know}$ category is taken from the total C categories for training, and the $C_{unknow}$ category is taken for testing. The query set contains N query pairs of <image, mask>, where the categories are the same as those in the support set. The model first extracts the feature knowledge of C categories from the support set and then performs the segmentation task on the query set using the extracted knowledge. Through continuous training and learning of different semantic classes, the model has a good generalization to new semantic classes. Finally, we put the model trained from the training set $D_{train}$ into the test set $D_{test}$ for segmentation performance evaluation.

### 3.2. Evaluation Indicators

In this paper, mIoU and binary-IoU are used as indicators to evaluate the performance of the model. The mIoU (Mean Intersection-over-Union) is the ratio obtained by computing the intersection and union of two sets of true and predicted values. Binary-IoU is to take all classes of objects as foreground and calculate the average IoU of foreground and background. We use mIoU and binary-IoU to evaluate the model performance comprehensively.

### 3.3. Method Overview

Different from most of the current few-shot segmentation methods, the method in this paper first extracts the guiding feature from the supported image through the feature extraction network to generate the prototypes. On this basis, we also fuse shallow features with deep features, generating prototypes with more detailed information. The query image is extracted by the feature extraction network to generate the query feature map. Similarly, we fuse the shallow feature and deep feature of the query image to generate a query feature map containing the detailed feature of the query image. Finally, we combine the feature maps of multiple prototypes and multiple query images to establish relationships by mixing similarities.

As shown in Figure 1, the model proposed in this paper performs the segmentation task as follows. First, a shared backbone network is used to extract the feature maps of support images and query images. Then, the feature maps of the support images are further processed by average masked pooling to obtain prototypes. Finally, the relationship between prototypes and query feature maps is established using our proposed hybrid similarity, as described in Section 3.5. To better measure the relationship between prototypes and query feature maps, multi-scale feature maps and multi-scale prototypes are constructed, and then relationships between prototypes and query feature maps are obtained at multiple scales, as described in Section 3.3. At the feature extraction stage, we adopt VGG-16 [14] as a shared backbone to extract deep features from support images and query images. At the same time, we fuse the shallow feature after the third convolution block with the deep feature after the last convolution block and then pass through the CBAM attention module to finally generate fused feature maps of support images and query images, as described in Section 3.4.

### 3.4. Multi-Scale and Multi-Prototypes Match

Currently, common few-shot semantic segmentation methods use single-scale prototypes and query features for similarity calculation. This method will lead to a rough match of the query feature because the prototypes feature cannot sufficiently represent the details of the plant disease feature. To address this challenge, a multi-scale and multi-prototypes match (MPM) method is proposed.

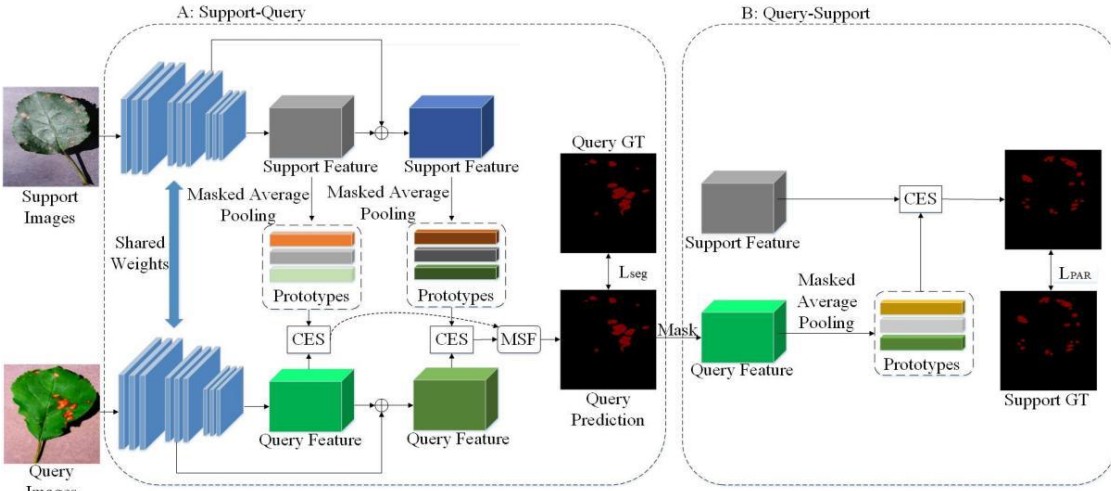

**Figure 1.** The overall pipeline of our proposed network(A: The model derives the diseased area of the query image from the support image with the diseased label; B: The model reversely deduces the diseased region of the support image by deriving the diseased region of the query image).

Suppose we obtain the support feature $F_{s1}$ and the query feature $F_{q1}$ from the supported image and the query image through the VGG-16 extraction network, respectively. Then, we fuse the feature after the third convolutional block of VGG-16 with the feature of the last convolutional block of VGG-16 to obtain the fusion feature $F_{s2}$ of support images and $F_{q2}$ of query images, respectively. In the same way, the feature of the second convolution block of VGG-16 and the feature of the fourth convolution block of VGG-16 can be fused so as to obtain the fusion feature $F_{s3}$ of the support images and the fusion feature $F_{q3}$ of the query images, respectively. Considering the efficiency of the model, here we take generating an additional support feature and generating an additional query feature as an example. First, we pass the feature $F_{s2}$ and $F_{q2}$ through the CBAM module so that the model pays more attention to the feature with a higher attention value in the training process. Next, we use a global pooling operation on the support feature $F_{s1}$ and $F_{s2}$ to generate prototypes $P_1$ and $P_2$. Finally, we calculate the similarity between multiple prototypes and multiple query feature maps.

### 3.5. CBAM Module

Although the fusion of shallow and deep features can obtain more detailed features of plant disease textures, it does not consider the differences between different pixel categories, channel features, and spatial features. Different feature learning weights affect the effect of plant disease segmentation. The introduction of the attention module can make the network model pay attention to the characteristics of plant disease areas during training and reduce unimportant learning weight coefficients, such as background area learning weight coefficients. So, introducing an attention module after the fusion of each shallow feature and deep feature allows the network to add different weights to a different feature.

The CBAM module [16] is a convolution-based attention mechanism module. Inspired by SENet [30], CBAM combines both channel attention and spatial attention, as shown in Figure 2. It can be clearly seen that CBAM is composed of CAM and SAM modules, which assign weights to channels and spaces, respectively.

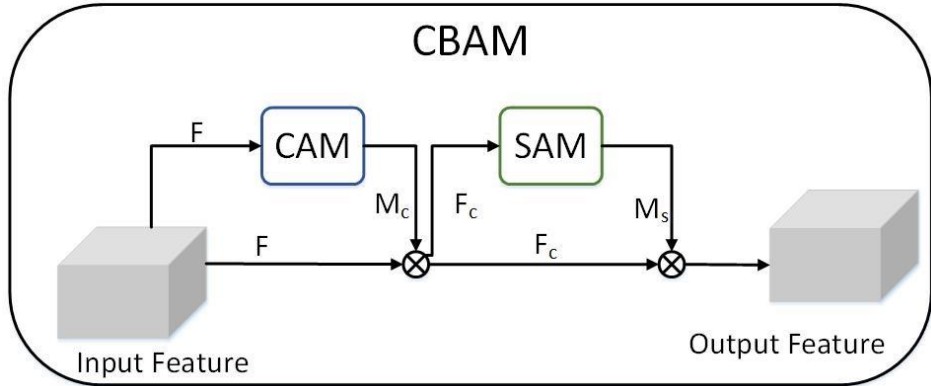

**Figure 2.** Structure diagram of CBAM (CAM stands for channel attention, SAM stands for spatial attention; *F*: input feature map; $M_c$: feature map after *F* passing through CAM; $F_c$: feature map after feature fusion of *F* and $M_c$; $M_s$: feature map after $F_c$ passing through SAM).

In the CAM module, max pooling and global average pooling operations are performed on the input feature map *F* (H × W × C) based on the height and width directions to obtain two (1 × 1 × C) feature maps. Next, the obtained feature map is respectively input to a multi-layer perceptron (*MLP*) for the pixel addition operation. Then the sigmoid activation function is applied, resulting in the final channel attention map $M_c$. Finally, the channel attention map $M_c$ and the input feature *F* are multiplied pixel by pixel to obtain the input feature $F_c$ required by the SAM module. The specific calculation is shown in the following Formula (1) [16]:

$$Mc(F) = \sigma(MLP(AvgPool(F)) + MLP(MaxPool(F))) \tag{1}$$

The SAM module takes the channel attention module output feature $F_c$ as the input feature. First, channel-based global max pooling and global average pooling are performed on $F_c$ to obtain two (H × W × 1) feature maps. Then, a 7 × 7 convolution is, respectively, acted on the feature map, which is the concatenation of two obtained feature maps along the channel dimension to reduce the number of channels to 1. Finally, the spatial attention map $M_s$ is obtained through the sigmoid activation function, and the attention map $M_s$ is multiplied pixel by pixel with the input feature $F_c$ of the module to obtain the final feature $F_s$ we need. The specific calculation process is shown in Formula (2) [16]:

$$M_s(F_c) = \sigma(f^{7 \times 7}([F^s_{avg}; F^s_{max}])) \tag{2}$$

*3.6. Hybrid Similarity*

Usually, in few-shot segmentation tasks, cosine similarity is used to establish the relationship between prototypes and query features. However, cosine similarity uses the cosine value of the angle between two vectors in the vector space as a measure of the difference between two individuals, which only distinguishes differences in direction and is not sensitive to absolute numerical values. In order to make up for this defect, the euclidean distance calculation was added to the original basis. The specific Formula (3) is as follows. Euclidean distance can reflect the difference between two individual numerical features, making up for the disadvantage that cosine similarity is not sensitive to the numerical value. The CES (cosine euclidean similarity) module is a new method for similarity calculation between prototypes and query features proposed in this paper. The principle is shown in Figure 3. In the three-dimensional space, we multiply the cosine value of the angle between the two vectors *A* and *B* and the euclidean distance between the vectors *A* and *B* by a certain scaling factor, and the sum of the two is used as a new way to establish the relationship between prototypes and query feature. The role of the scale factor is to balance the effects of cosine similarity and euclidean distance on the calculation of the difference between two vectors. When establishing the relationship between the prototypes and the

query feature maps, the CES module can not only consider the similarity of the two vectors in the spatial direction but also pay attention to the similarity of the two vectors in the spatial distance.

$$Similaryty(A, B) = \frac{\sum\limits_{i=1}^{n}(A_i \times B_i)}{\sqrt{\sum\limits_{i=1}^{n}A_i^2} \times \sqrt{\sum\limits_{i=1}^{n}B_i^2}} \times Factor + \frac{1}{\sqrt{\sum\limits_{i=1}^{n}(A_i - B_i)^2}} \times (1 - Factor) \quad (3)$$

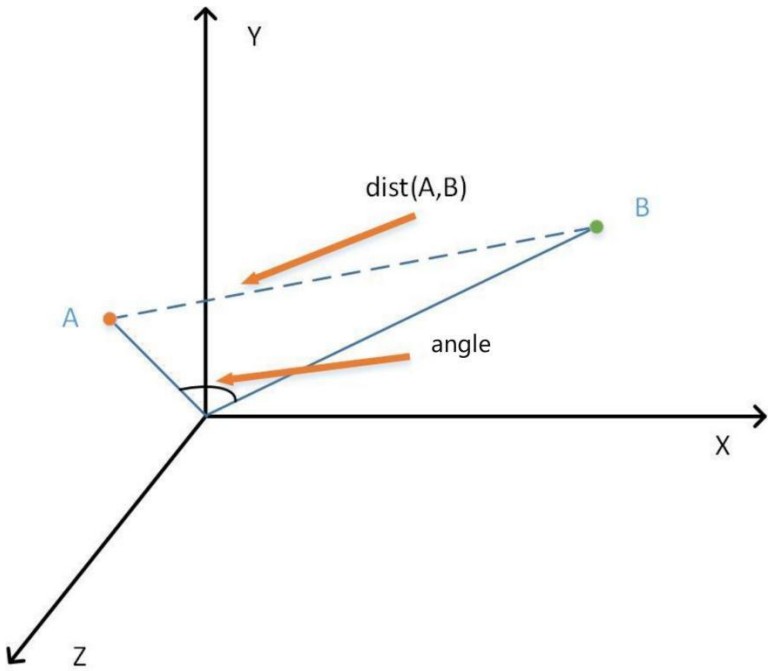

**Figure 3.** Schematic explanation of the CES module (dist means to calculate the Euclidean distance between vectors *A* and *B*; angle means the angle between the two vectors).

*3.7. Loss Function*

As shown in Figure 1, our model is trained according to two processes, which are Support-Query and Query-Support, respectively. Specifically, the process of Support-Query is to learn knowledge from the annotated support set, establish the relationship with the query image, and then predict the query image to obtain the segmentation result. We calculate the loss between the obtained segmentation result and the real label of the query image, as shown in Formula (4) [15]. In contrast, the process of Query-Support, which is only performed during training, is to flow query information to the support set. Average pool operation is employed to the query feature to obtain another set of prototypes, which are used to match with the support feature maps to obtain the support prediction results. Next, this paper calculates the loss of the prediction result according to Formula (5) [15] and then returns to the training process to adjust the weight.

$$L_{seg} = -\frac{1}{N} \sum_{x,y} \sum_{p_j \in \rho} \left[ M_q^{(x,y)} = j \right] \log \widetilde{M}_{q;j}^{(x,y)} \quad (4)$$

$$L_{PAR} = -\frac{1}{CKN} \sum_{c,k,x,y} \sum_{p_j \in \rho} \left[ M_q^{(x,y)} = j \right] \log \widetilde{M}_{q;j}^{(x,y)} \quad (5)$$

where $M_q$ is the ground truth segmentation mask of the query image, $(x,y)$ denotes the index of the spatial location, C indicates that the current processing is the one in a set of support images, K represents the current category and N is the total number of spatial locations.

## 4. Experiments

### 4.1. Experimental Setup

Implementation details: We adopt VGG-16 as the shared feature extraction network. The size of the input image is (417, 417), and an SGD optimizer is adopted to realize the end-to-end training process. The hyperparameters of the training phase, such as learning rate, training iterations, weight decay, momentum, and batch size, are set to 0.001, 30,000, 0.0005, 0.9, 1, respectively.

Dataset: In order to meet the plant disease few-shot semantic segmentation task, we created a dataset PDID that fully meets the requirements of the task. It contains 10 plant diseases, a total of 963 images, and category information. PASCAL-5$^i$ [31] is a dataset for evaluating few-shot segmentation. The dataset is subdivided into four parts, each containing five categories, with a total of twenty categories. One section contains annotated samples from five classes to evaluate few-shot learning methods. The remaining fifteen categories are used for training. According to the format of PASCAL-5$^i$, we select five categories from the PDID dataset as the train dataset $D_{train}$ and the remaining categories as the test dataset $D_{test}$, which are denoted by PDID-5$^0$ and PDID-5$^1$. The specific categories are shown in Table 1. In order to simulate the real lighting environment and shooting angle, rotation, and color jitter, data augmentation operations are used to expand the dataset to 9630 images, which then is divided into a training set, validation set, and test set according to the ratio of 8:1:1. In the K-shot setting, we take K + 1 images with the same class annotation labels from the training set $D_{train}$, the K images and annotation labels are input to the model as support images for analysis. The remaining image is input into the network model as a query image, and its annotation label is used as the ground truth for loss calculation.

**Table 1.** Category names in dataset PDID-5$^i$.

|  | C1 | C2 | C3 | C4 | C5 |
|---|---|---|---|---|---|
| **PDID-5$^0$** | Apple Frogeye Spot | Apple Scab | Grape Black Measles Fungus | Grape Black Rot Fungus | Grape Leaf Blight Fungus |
|  | **C6** | **C7** | **C8** | **C9** | **C10** |
| **PDID-5$^1$** | Peach Bacterial Spot | Tomato Early Blight Fungus | Tomato Late Blight Water Mold | Tomato Leaf Mold Fungus | Tomato Septoria Leaf Spot Fungus |

### 4.2. Experimental Results and Discussions

#### 4.2.1. Validation of Proposed Model

To validate the effectiveness of the proposed model, we have compared the proposed model with other methods on the PDID dataset. In this experiment, VGG-16 is used as the backbone network, and the comparison results are shown in Table 2. In Table 2, 1-shot means that our support image is one, and 1-way means that we only extract one of the 10 categories.

**Table 2.** Results of 1-way 1-shot segmentation on PDID-5$^i$ dataset using mean-IoU metric.

| Method | 1-Way 1-Shot | | | Params |
|---|---|---|---|---|
|  | Split-0 | Split-1 | Mean |  |
| PANet | 43.1 | 34.5 | 38.8 | 14.7 M |
| Ours | 45.4 | 35.6 | 40.5 | 16.5 M |

The results show that the mIoU of our proposed model is 2.3% and 1.1% higher than that of PANet on Split-0 and on Split-1, respectively, and the average of mIoU on Split-0 and on Split-1 is increased by 1.7%. So we can conclude that our model is significantly more

outstanding than other methods and is more suitable for the segmentation of few-shot plant disease images. In addition, more performance enhancement in Split-0 is obtained than that in Split-1, which shows that Split-0 is easier to transfer knowledge from the base class to the new class.

### 4.2.2. Performance Comparison of Different Types of Diseases

To further analyze the performance of different categories of plant diseases under few-shot segmentation, we list the mIoU results for each category under the 1-shot setting, as shown in Table 3. According to the table, we can clearly see that almost all categories have an excellent performance. In particular, C8, C1, C3, and C4 are improved the most compared to PANet among all categories, which is 3.3%, 2.6%, 2.5%, and 2.5%, respectively.

**Table 3.** Performance comparison of different classes on PDID-5$^i$ dataset.

| Method | C1 | C2 | C3 | C4 | C5 | C6 | C7 | C8 | C9 | C10 |
|---|---|---|---|---|---|---|---|---|---|---|
| PANet | 57.3 | 35.3 | 42 | 54.4 | 26.7 | 40.6 | 52.4 | 29.4 | 23 | 27 |
| Ours | 59.9 | 36.6 | 44.5 | 56.9 | 29.2 | 41.3 | 52.2 | 32.7 | 23.6 | 28.2 |

### 4.2.3. Qualitative Analysis

In order to show the effectiveness of our method more vividly, we have visualized the excellent segmentation results, and the qualitative results are shown in Figure 4, which can demonstrate the strong segmentation ability of our network for plant diseases. Compared with the ground-truth labels, we can clearly find that the segmentation effect of our model is not much different from that of the real disease area, which also shows that our model can extract object feature well from the support image and then be extended to the query image. In particular, our proposed model can better identify dense, small-target plant disease areas. This phenomenon can be further explained by the fact that MPM adds multiple prototypes and multiple query features to match each other, which can make up for some details lost with the increase of network depth. However, for some extremely small diseases, our model has some missing results, as shown in Figure 5. The main reason for this phenomenon is that extremely small diseases have less feature information and are easily lost in the process of feature extraction.

### 4.2.4. Training Loss Function

As shown in Figure 6, after 30,000 iterations, the loss functions $L_{seg}$ and $L_{PAR}$ both decrease rapidly around 1000 epochs, then slow down as the epochs increase, almost leveling off at 30,000 epochs. Among them, $L_{seg}$ represents the loss of the segmentation result of the query image at the support–query stage, and. $L_{PAR}$ calculates the segmentation loss of support images at the query–support stage, which makes the support prototypes and query image prototypes aligned with each other.

### *4.3. Ablation Studies*

To further verify the effectiveness of our designed module, we conduct ablation experiments on the dataset PDID-5$^0$ with a 1-shot setting. First, we study the effect of different scale coefficients in the CES module on the model performance and use the adjustment scale factor to distribute the computational shares of cosine similarity and euclidean distance. CES with different scale coefficients has different effects on computing prototypes and query feature maps. Among the plant leaf disease feature extracted by the feature extraction network, it may be more sensitive to the angle between vectors when calculating similarity, or it may be more sensitive to the euclidean distance between vectors. So we find an optimal scale factor by constantly adjusting different scale coefficients (cosine similarity and euclidean distance). As shown in Table 4, the optimal performance occurs at a ratio of 9:1 (share of cosine similarity: share of euclidean distance) with the mIoU value of 45.4. After that, the model performance decreases as the share of cosine similarity

decreases (except for a slight improvement at 7:3 compared to 8:2). In summary, we adopt the CES module with a ratio of 9:1 for calculating the similarity between prototypes and query feature maps in our model.

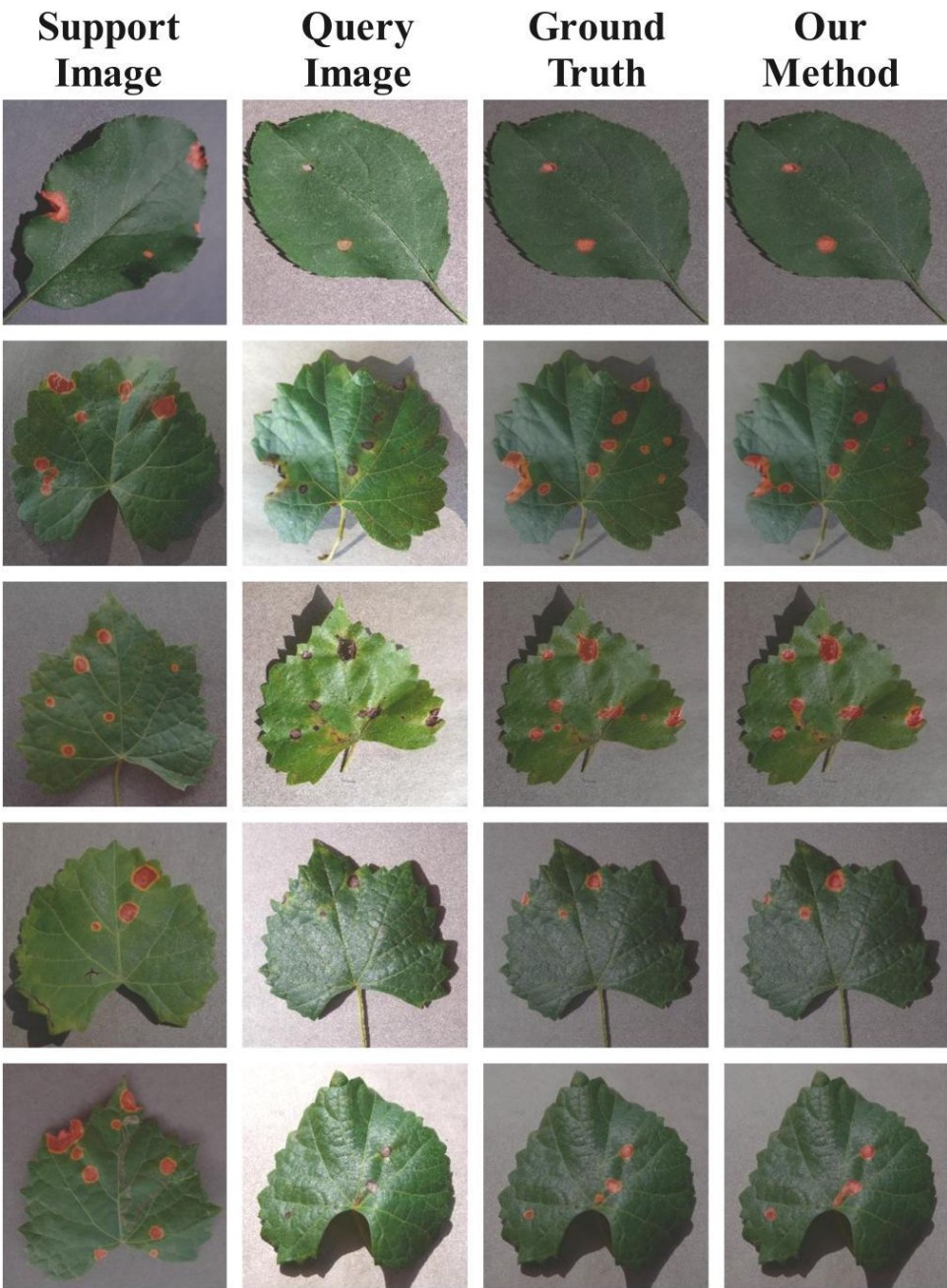

**Figure 4.** Qualitative results of our model in 1-way 1-shot segmentation on PDID-5[i].

**Table 4.** Ablation experiments on different components.

| Ratio | 9:1 | 8:2 | 7:3 | 6:4 | 5:5 | 4:6 | 3:7 | 2:8 | 1:9 |
|---|---|---|---|---|---|---|---|---|---|
| mIoU | **45.4** | 44.1 | 44.5 | 44.3 | 42.7 | 42.3 | 41.6 | 40.5 | 39.6 |
| binary-mIoU | **69.2** | 68.3 | 68.4 | 67.9 | 66.7 | 66.5 | 65.5 | 64.2 | 63.4 |

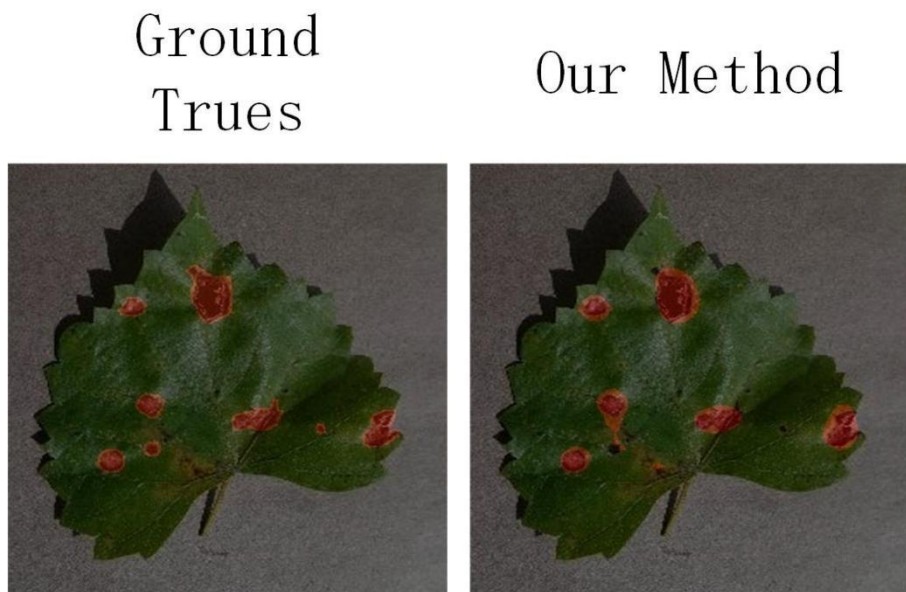

**Figure 5.** Segmentation effect error.

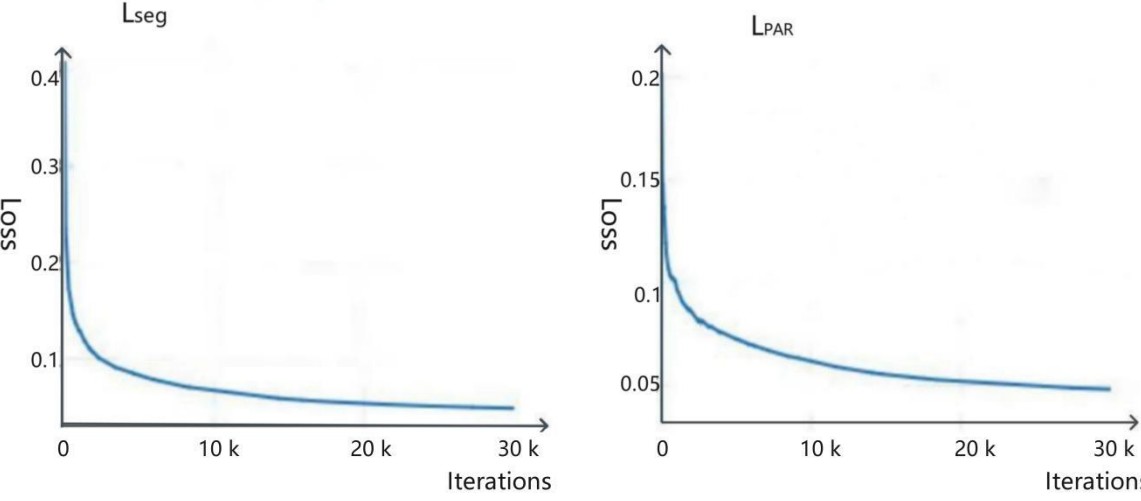

**Figure 6.** The training loss of the model.

In order to verify the impact of different ways of establishing the relationship between the prototypes and the query feature maps on the performance and efficiency of the model, we conducted seven sets of experiments, as shown in Table 5. We name the initial prototypes and initial query feature maps P and Q, respectively, and the new prototypes and query feature maps obtained through multi-scale feature fusion are P', Q'. Way 1 represents the relationship between the prototype P and the query feature map Q, and the new prototype P' and the new query feature map Q'. Similarly, way 2 represents the establishment of the relationship between P-Q' and P'-Q; way 3 represents the establishment of the relationship between P-Q, P-Q', and P'-Q'; way 4 represents the establishment of the relationship between P-Q, P-Q', and P' -Q relationship establishment; way 5 represents the relationship establishment of P-Q, P'-Q, P'-Q'; way 6 represents the relationship establishment of P-Q', P'-Q, P'-Q'; way 7 represents The relationship of P-Q, P-Q', P'-Q, P'-Q' is established. From Table 5, it can be clearly seen that the relationship establishment method of way 1 is more suitable for our network model and achieves the best performance.

**Table 5.** Multiple prototypes and multiple query feature maps to establish relationships.

| Way | mIoU | Binary-mIoU |
| --- | --- | --- |
| 1 | **45.4** | **69.2** |
| 2 | 37.7 | 64.1 |
| 3 | 42.6 | 66.5 |
| 4 | 42.9 | 67.4 |
| 5 | 44 | 68.5 |
| 6 | 36.8 | 63.8 |
| 7 | 42.1 | 66.5 |

In addition, we further verify the impact of the designed modules on the model performance. As shown in Table 6, a 0.6% mIoU improvement is obtained by adding our designed CES module model. By adding multi-scale and multi-prototypes match, the mIoU of the model is increased by 0.9%, and the binary_mIoU is increased by 0.9%. When the two modules were jointly combined, the mIoU of our few-shot segmentation network was up to 45.4%, an improvement of 2.3% compared to 43.1%.

**Table 6.** Ablation experiments on different components.

| CES | MPM | mIoU | Binary_mIoU |
| --- | --- | --- | --- |
|  |  | 43.1 | 67.9 |
| √ |  | 43.7 | 68.5 |
|  | √ | 44 | 68.8 |
| √ | √ | **45.4** | **69.2** |

During the test, in order to prevent the error caused by different detections each time, this paper conducts five tests and finally takes the average value. As shown in Figure 7, the model proposed in this paper is higher than the initial network in each test, so the conclusion that our network has a good segmentation ability for plant diseases can be drawn.

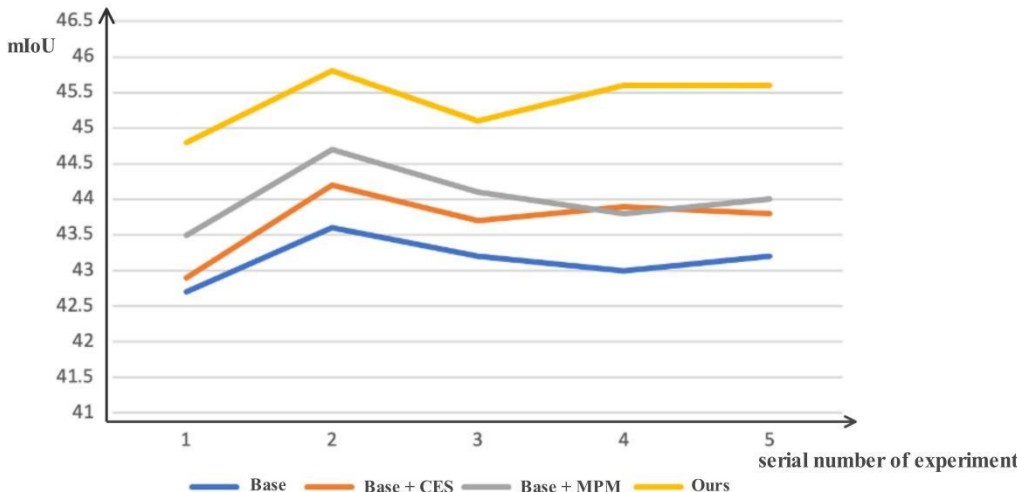

**Figure 7.** Five test comparisons of different models.

## 5. Conclusions

We propose a novel multi-scale and multi-prototype matching few-shot segmentation network. First, the MPM structure in the network obtains prototype and query features at two different scales through multi-scale feature fusion, which can enrich the relationship between leaf disease prototypes and query features, thereby improving the accuracy of the model to identify disease areas. Furthermore, this paper proposes a new method

for establishing the relationship between prototype and query feature maps, which takes into account the different computations of angles and distances between the two vectors. Experiments on the scale coefficient show that the performance of this method can reach the best according to a ratio of 9:1 (cosine similarity: euclidean distance), and the mIoU reaches 40.5%. Extensive experiments are carried out on the plant disease dataset PDID-5[i]. From the experimental results, it can be seen that the mIoU of our proposed network model is improved by 1.7% compared with that of the original PANet, which proves that our model has an excellent performance in segmenting plant leaf disease-infected regions. It provides the possibility for the prevention and control of early plant infection diseases and can reduce economic losses and increase yield.

**Author Contributions:** W.H. conceived the paper, designed and conducted experiments, and wrote the article. W.Y. provided guidance for thesis innovation and guided thesis revision. Z.Y. verified experimental design and analyzed data. L.X. provided constructive comments on the research and revised the paper. All authors have read and agreed to the published version of the manuscript.

**Funding:** This research was funded by the Natural Science Foundation of Jiangxi Province, grant number 20212BAB212005; Science and Technology Project of Jiangxi Provincial Department of Education, grant number GJJ190217; Open Project of State Key Laboratory, University of Zhejiang number A2029.

**Data Availability Statement:** Dataset is at https://github.com/0blackcrow0/PDID/tree/master (accessed on 29 July 2022).

**Acknowledgments:** The authors would like to thank the anonymous reviewers for their critical comments and suggestions for improving the manuscript.

**Conflicts of Interest:** The authors declare no conflict of interest.

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
