# Peer review of "Multi-Scale and Multi-Match for Few-Shot Plant Disease Image Semantic Segmentation"

_agronomy, doi:10.3390/agronomy12112847_

Round 1
Reviewer 1 Report
Dear authors, I reviewed the paper entitled “Multi-scale and multi-match for few-shot plant disease image semantic segmentation” by Wenji Yang et al. The paper deals with a novel procedure for semantic segmentation applied to few-shot plant diseases. This new model claims to be multi-scale and a multi-prototypes match. A CES module is also designed for solving the shortcoming of traditional cosine similarity and lack of spatial distance awareness. The procedure is well explained and its superiority regarding other methods is demonstrated with practical experiments.
The paper has many acronyms and all of them must be defined for the non-expert reader.
The English must also be reviewed by a native speaker.
The paper must be restructured. According to the instruction to authors of Agronomy, the research paper must have a separate result and discussion sections. According to this, include a results section with the results of the experiment and a discussion section where in addition to pros and contras of your new method, a comparison with other methods and studies from other authors must be included.
The introduction should define clearly the objectives of the paper. It defines the main contributions but, in addition, the objective should be clearly defined. In addition, more references to other segmentation methods and practical cases could be included.
Add a reference for PlantVillage
Add references for equations (1) to (5)
In section 3.5, a scaling factor is defined. It should be clear which this value is or how it is selected for a certain segmentation. Also in the experiment, it should be clear how it is selected.
In Figure 3, you should replace “cos” by “angle”. You are indicating the angle, not the cos.
The section “”Loss function” is section 3.6
Indicate what C and K mean in equation (5)
The evaluation indicators used in the experimental analysis should be defined in the methodology.
In Figure 4, indicate what GT means and also replace Our with Our method, for example.
In Figure 5, if you use the same scale for the vertical axis in both plots, the difference would be more visually clear.
Author Response
Manuscript ID: agronomy-1964305
Manuscript Title: Multi-scale and Multi-match for Few-Shot Plant Disease Image Semantic Segmentation
We submitted the revised manuscript; the revisions were carried out based on the editor and each reviewer’s comments. We thank the editor and all reviewers for their valuable comments and suggestions. The following summarizes our point-by-point responses to comments and suggestions.
---------------------------------------------
<< Author's Reply to the Review Report (Reviewer 1)>>
- The paper has many acronyms and all of them must be defined for the non-expert reader.
Thank you for your suggestions on this article. First of all, We are sorry that we have used inappropriate acronyms. Now we have added their definition when they first appeared in the paper.
- The English must also be reviewed by a native speaker.
Thank you very much for this suggestion, the grammar has now been corrected throughout the paper.
- The paper must be restructured. According to the instruction to authors of Agronomy, the research paper must have a separate result and discussion sections. According to this, include a results section with the results of the experiment and a discussion section where in addition to pros and contras of your new method, a comparison with other methods and studies from other authors must be included.
We appreciate for the constructive comments provided to help improve the structure of the paper. According to your suggestions and the guidance of the agronomy authors, we now have included the experimental results and discussion parts in detail. Starting with the first point of our improvement (proposed MPM module), the analysis and discussion of the experimental results are carried out in Section 4.3. The role of the MPM module is to generate multiple prototypes through multi-scale fusion. These prototypes contain more details and texture characteristics of plant leaf diseases. After adding MPM to the original network structure, the mIoU is increased by 0.9%, effectively proving the effectiveness of the module.The second content of the experiment 4.3 (mixed similarity) is to discuss the influence of different proportion coefficients on the calculation of mixed similarity. Finally, An optimal scale factor is obtained through experiments, which is 9:1 (cosine similarity: Euclidean distance), and the mixed similarity improves the model performance by 0.6%.
In Section 4.2.1, we verify the proposed model and discuss the advantages of our method to improve performance. In Section 4.2.2, this paper compares the segmentation performance of the proposed method for different kinds of diseases. In Section 4.2.3, this paper analyzes the performance of the model through visual methods, and discusses some shortcomings of the model.
For details, please refer to page 9 line 333-428.
- The introduction should define clearly the objectives of the paper. It defines the main contributions but, in addition, the objective should be clearly defined. In addition, more references to other segmentation methods and practical cases could be included.
Thank you for your suggestions on the introduction of this article, according to your suggestions, we have modified the introduction. In the introduction, we added the following:In this paper, semantic segmentation is performed for early disease images of plant leaves, and the diseased areas in leaves are drawn, which provides a new solution for plant disease control.
For details, please refer to page 2 line 65-67.
- Add references for equations (1) to (5)
Thank you for your comments on formulas (1) to (5) in this article. The corresponding references have been added to the formula (1) (2) (4) (5) mentioned in this article. The formula (3) is proposed in this article, so it is not added. Quote.
formula (1) corresponding to references [9].
formula (2) corresponding to references [9].
formula (4) corresponding to references [8].
formula (5) corresponding to references [8].
- In section 3.5, a scaling factor is defined. It should be clear which this value is or how it is selected for a certain segmentation. Also in the experiment, it should be clear how it is selected.
I am glad that you provide these valuable suggestions for this article. We have explained the role of the scale factor in Section 3.6, The role of the scale factor is to balance the effects of cosine similarity and euclidean distance on the calculation of the difference between two vectors. For details, please refer to page 7 line 281-283.
“Why we add this scale factor, and how to choose it” are given in Experiment 4.3. Among the plant leaf disease feature extracted by the feature extraction network, it may be more sensitive to the angle between vectors when calculating similarity, or it may be more sensitive to the euclidean distance between vectors. So we find an optimal scale factor by constantly adjusting different scale coefficients (cosine similarity and euclidean distance). For details, please refer to page 11 line 382-386.
- In Figure 3, you should replace “cos” by “angle”. You are indicating the angle, not the cos.
Thanks you extremely to the article revision. Based on your suggestions, The angle is indeed more appropriate than the cos in Figure 3. We have revised Figure 3, replacing “cos” with “angle”.
For details, please refer to page 7 line 288.
- The section “”Loss function” is section 3.6
Thanks for pointing out this error. According to the layout of the full article, it has been revised to 3.7.
For details, please refer to page 7 line 290.
- Indicate what C and K mean in equation (5)
We have explained C, K below Equation 5; C indicates the index of the support images, K represents the current image category.
For details, please refer to page 8 line 306-307.
- The evaluation indicators used in the experimental analysis should be defined in the methodology.
Thank you very much for this suggestion. After careful consideration, we also feel that the evaluation indicators are more appropriate to be defined in the method. We have put the evaluation indicators in Section 3.2 of the method.
For details, please refer to page 4 line 178-184.
- In Figure 4, indicate what GT means and also replace Our with Our method, for example.
Thank you very much for your careful review of this article. GT is the abbreviation of Ground Truth. It represents the true value of the leaf disease label, that is, the actual diseased area on the leaf. We also changed "Our" to "Our Method".
For details, please refer to page 10 line 365.
- In Figure 5, if you use the same scale for the vertical axis in both plots, the difference would be more visually clear.
I am glad that you provide this valuable suggestions for our article. The two loss function curves in Figure 6 represent Lseg and LPar, respectively. They are not used for comparison, but to see if the two curves have flattened out. If the two loss function curves tend to be flat, it means that our model training has been fitted; if the curves are not flat, it means that the model has not converged and needs to continue training.
For details, please refer to page 11 line 383
---------------------end---------------------
We appreciate very much for your kind suggestion and make all revisions in the updated manuscript. Thank you!

Reviewer 2 Report
Dear Authors,
you wrote an impressive article about the enhancement of cnn and how to overcome the lag of missing training data for the cnn. Whereby the databases for street signs and common objects are growing, there are just a few available for agricultural purpose. Solutions for automatic segmentation and training with a few references are highly appreciated and of high research impact.
What I`m not so sure is the fact that most of the agronomy readers are not ai experts, and you overstrain the readers with expert knowledge. By this, it is tough to read, and additionally your references are from other areas and not agronomy.
Is this really already all common knowledge?:
“multi-scale and multi-prototypes match (MPM); masked average pooling; low-scale features and high-scale features; cosine similarity; spatial distance; CES module; PDID-5i dataset; VGG16; sigmoid activation function; back-bone network; shallow features and deep features; feature fusion structures;
Please explain all abbreviations at the first time in the dok.: CES, PDID-5i, …
Line 35: Sentence! “ such as” you mean “using (for sem. Seg.)”
Line 50: VGG16 describe, maybe convolutional neural network model proposed by K. Simonyan and A. Zisserman from the University of Oxford…
Line 59: Sentence! simply match the; matching?
Line 64: the high-scale support feature? What does high scale mean, and why is it the feature? Is there just one specific existing? You often mix feature and features, it is very confusing.
Line 78: the hybrid similarity calculation? What does that mean? Do you mean This or a hybrid …?
Line 80: What is “a certain coefficient”? Please be more precise.
Line 84: CBAM attention mechanism; again it is not nice that I always must go to the literature to find out your abbreviations.
Line 85: what are:” shallow features and deep features”?
Line 90: PlantVillage? Explain that it is a dataset!
Line 96: measures?
Line 105: “we carefully constructed” this is no scientific terminus and not understandable! Line 90 you mention to use plantvillage dataset, so what is going on?
Line 116: “some other works” whose other works?
Line 117: What are “feature fusion structures”?
And so on…
Line 192 Figure 1, is it really just one feature or do you mean features?
Line 283: M q àMq
Line 294: “PASCAL-294 5i,” What is this?
Line 330: MPM?
Line 341 Table 2: again what does that mean?: Result of 1-way 1-shot ? segmentation on PDID-5i dataset using mean-IoU metric
Line 344 Figure 4: I can`t see any difference or result, please use different color for the computational result!!!!!!!!!!!!!!!!!!!!!!! Mark the result or zoom in!
Line 346 Figure 5 Image quality! Where are the xy axis? Why is it in gray…
Line 348 – 384 What is it good for and what is the impressive research outcome?
Line 389 Figure 6 Quality! XY-axis?
Line 391: Poor short conclusion. Even here you not mention what are 4.3 results are good for!
Author Response
Manuscript ID: agronomy-1964305
Manuscript Title: Multi-scale and Multi-match for Few-Shot Plant Disease Image Semantic Segmentation
We submitted the revised manuscript; the revisions were carried out based on the editor and each reviewer’s comments. We thank the editor and all reviewers for their valuable comments and suggestions. The following summarizes our point-by-point responses to comments and suggestions.
---------------------------------------------
<< Author's Reply to the Review Report (Reviewer 2)>>
- What I`m not so sure is the fact that most of the agronomy readers are not ai experts, and you overstrain the readers with expert knowledge. By this, it is tough to read, and additionally your references are from other areas and not agronomy.
For this suggestion ,I would like to make the following explanation: First of all, I am very sorry. The artificial intelligence technology mentioned in this paper has brought some difficulties to agricultural readers. Some AI knowledge is explained in the article. Secondly, this paper is to apply artificial intelligence technology for the detection of plant diseases, which can provide a new solution for the prevention and control of plant leaf diseases.
- Is this really already all common knowledge?:
“multi-scale and multi-prototypes match (MPM); masked average pooling; low-scale features and high-scale features; cosine similarity; spatial distance; CES module; PDID-5i dataset; VGG16; sigmoid activation function; back-bone network; shallow features and deep features; feature fusion structures;
This suggestion is very valuable, I want to explain to this suggestion. In the field of AI, these words have been considered as common knowledge, in this article in order to make more agronomy readers understand the new field applied to agriculture. We have described these proper nouns using the most accessible narrative possible.
What we hope more is to combine artificial intelligence technology with agriculture to provide more methods for the agricultural field, such as the prevention of early diseases and pests mentioned in this article.
- Please explain all abbreviations at the first time in the dok.: CES, PDID-5i, …
Thank you very much for your valuable suggestions, we have supplemented some abbreviated words throughout the article.
CES has been changed to CES(cosine euclidean similarity). For details, please refer to page 1 line 28. The CES (cosine euclidean similarity) module is a new method for similarity calculation between prototypes and query feature proposed in this paper. For details, please refer to page 6-7 line 276-277.
We have described the PDID-5i dataset in detail: This paper constructs a plant disease dataset(PDID-5i) containing ten different categories, which are then annotated at pixel level, and conducts experiments on the dataset to verify the effectiveness of our network. For details, please refer to page 2 line 97-99.
- Line 35: Sentence! “ such as” you mean “using (for sem. Seg.)”
First of all, thank you very much for your valuable comments. Through this comment, I feel your carefulness and attention to the research content of this paper. According to your comments, we carefully analyze and believe that the use of "using" is more in line with the meaning of the article and the revised sentence is as follows: semantic segmentation using FCN [1], UNet [2], SegNet [3], Deeplab [4], ASPP [5] has become one of the main technologies of agricultural intelligence.
For details, please refer to page 1 line 36.
- Line 50: VGG16 describe, maybe convolutional neural network model proposed by K. Simonyan and A. Zisserman from the University of Oxford…
Thank you for your point of view, according to the newly added reference [32], it shows that VGG16 is indeed a convolutional neural network model proposed by Simonyan K. Its role is to extract features. In the network model architecture we propose in this paper, the VGG16 is used to extract the guiding features of the support image.
- Line 59: Sentence! simply match the; matching?
Thank you very much for your valuable comments on this article, the simply match here does have a grammatical error. We thank you again for your suggestions for this article. We have corrected "match" to "matching" in this article.
For details, please refer to page 2 line 61.
- Line 64: the high-scale support feature? What does high scale mean, and why is it the feature? Is there just one specific existing? You often mix feature and features, it is very confusing.
Thank you very much for pointing out the shortcomings of this article. First of all, the high-scale support feature here refers to the feature maps in the higher layers of the neural network. At the same time, we have modified “feature” and “features” appeared throughout the paper into “feature”.
- Line 78: the hybrid similarity calculation? What does that mean? Do you mean This or a hybrid …?
The hybrid similarity we mentioned here refers to a new similarity calculation method that we propose to fuse cosine similarity and Euclidean distance calculation, which can focus on the difference of angle and distance at the same time.
- Line 80: What is “a certain coefficient”? Please be more precise.
Thank you very much for your comment, we have added the specific ratio to the article. The details are as follows:
Therefore, a hybrid similarity calculation is adopted, which calculates the euclidean distance and the cosine similarity of the two vectors, and then a weighted sum is performed according to 9:1 (the cosine similarity: the euclidean distance). In this way, the method can obtain more accurate similarity maps. For details, please refer to page 2 line 84-85.
- Line 84: CBAM attention mechanism; again it is not nice that I always must go to the literature to find out your abbreviations.
I'm very sorry that my negligence did not bring you a good reading experience.Thank you for your comments. We have supplemented the English abbreviations that appear at the first time in the article. At the same time, we have revised the similar situation in the full article.
For details, please refer to page 2 line 89-90.
- Line 85: what are:” shallow features and deep features”?
Features of different scales are actually pictures of different resolutions.
Shallow feature: the feature extracted by the shallow network are relatively close to the input, and it is a high-resolution image that contains more pixel information. Some fine-grained information is some color, texture, edge, and corner information of the image.
Deep feature: the feature extracted by the deep network is closer to the output, which is a low-resolution picture, showing some coarse-grained information, including more abstract information, that is, semantic information.
- Line 90: PlantVillage? Explain that it is a dataset!
Taking your advice, we have modified this non-standard statement. The data set mentioned in this sentence is the PDID dataset we constructed, which is used for the subsequent experiments in this paper. To avoid misunderstanding of the experimental dataset, we delete the PlantVillage dataset.
For details, please refer to page 2 line 95.
- Line 96: measures?
Thank you for your suggestion. First of all, I am sorry for the vocabulary we used, which is not enough to express the description of similarity calculation in this study clearly, and secondly, in order to make the expression of the article clearer and more intuitive, “measures” has been revised to “measure method”. For details, please refer to page 3 line 104-105. If this statement is not very clear, we kindly ask for your valuable comments on revisions, and thank you again for your comments on this statement.
- Line 105: “we carefully constructed” this is no scientific terminus and not understandable! Line 90 you mention to use plantvillage dataset, so what is going on?
Thank you very much for your valuable comments. We have revised this non-standard expression. At the same time, the data set mentioned in this sentence is the PDID data set we constructed, which is used for the subsequent experiments in this paper. In order to avoid readers' misunderstanding of the experimental dataset, we delete the PlantVillage dataset.
For details, please refer to page 2 line 97.
- Line 116: “some other works” whose other works?
About this suggestion, we have revised this non-standard expression. Some other works mentioned in the original article refers to Lin et al. [16] using a feature fusion structure to improve the accuracy. However, as you propose, this description creates certain problems. We also have removed expressions which can be misunderstood by people.
For details, please refer to page 3 line 123-125.
- Line 117: What are “feature fusion structures”?
I'm glad for your comment on this article. The function of "feature fusion structures" mentioned in this article is to fuse high-resolution images with low-resolution ones, because high-resolution images contain more textures of plant diseases, color and other features, while the low-resolution images obtained as the depth of the convolutional network deepens contain the semantic features of the disease. Fusion of the two can make the model locate the disease area more accurately.
- Line 192 Figure 1, is it really just one feature or do you mean features?
I'm glad you point out this problem, feature in convolutional neural networks is nouns with abstract meanings, not a plural of nouns.
All features has been changed to feature.
For details, please refer to page 2 line 46, line 49, line 51, line 52,etc.
- Line 283: M qàMq
Thanks for pointing out format error. We have changed “Mq” to “Mq” in the article. At the same time, I have checked the similar problem in the full article.
For details, please refer to page 8 line 305-307.
- Line 294: “PASCAL-294 5i,” What is this?
The "PASCAL-5i" mentioned in the paper is a public dataset suitable for few shot semantic segmentation. Based on the format of this dataset, we make a Plant Disease Dataset (PDID) suitable for few shot semantic segmentation. To make it easier for agronomy experts to understand, we have supplemented the description of "PASCAL-5i" in the article.
For details, please refer to page 8 line 318.
- Line 330: MPM?
Sorry again for the inconvenience caused by the abbreviation, the MPM we mentioned here is a new method proposed in Section 3.4 — multi-scale and multi-prototypes match.
For details, please refer to page 5 line 211-216.
- Line 341 Table 2: again what does that mean?: Result of 1-way 1-shot ? segmentation on PDID-5i dataset using mean-IoU metric
1-shot in Table 2 means that our support image is one, and 1-way means that we only extract one of the 10 categories.
- Line 344 Figure 4: I can`t see any difference or result, please use different color for the computational result!!!!!!!!!!!!!!!!!!!!!!! Mark the result or zoom in!
Figure 4 is qualitative results of our model in 1-way 1-shot segmentation on PDID-5i. The red area in the figure has already marked the result. The result of the ground truth column is a result that our model wants to achieve, and the result of our method column is the plant disease area predicted by our model. The closer the two area are, the better the performance of our model is.
- Line 346 Figure 5 Image quality! Where are the xy axis? Why is it in gray…
Thank you very much for pointing out the shortcomings of Figure 5. We added the missing xy axis in Figure 6 and changed the gray part.
For details, please refer to page 11 line 383.
- Line 348 – 384 What is it good for and what is the impressive research outcome?
We set up an ablation experiment in this paper to prove that putting each module into the model is beneficial to improve the segmentation accuracy, and also conduct experiments on the proportional coefficient of the mixed similarity to verify that we are using the best coefficient. The experimental results show that the addition of MPM module increases the mIoU by 0.9%, and the addition of CES increases the mIoU by 0.6%. When two modules are added to the model at the same time, the performance of the model is improved by 2.3%.
For details, please refer to page 13 line 417-422(Table 6).
- Line 389 Figure 6 Quality! XY-axis?
Thanks for your suggestion, we have added the XY-axis in Figure 6 and explained what it means.The X-axis is serial number of experiment, the Y-axis is mIoU.
For details, please refer to page 13 line 430 (Figure 7).
- Line 391: Poor short conclusion. Even here you not mention what are 4.3 results are good for!
Thank you for your suggestions on conclusion. For the conclusion part, we give a new description according to the research purpose and experimental results.We also describes the shortcomings of this study and the future research direction.
We propose a novel multi-scale and multi-prototype matching few-shot segmentation network. First, the MPM structure in the network obtains prototype and query feature at two different scales through multi-scale feature fusion, which can enrich the relationship between leaf disease prototypes and query feature, thereby improving the accuracy of the model to identify disease areas. Furthermore, this paper proposes a new method for establishing the relationship between prototype and query feature maps, which takes into account the difference computation of angles and distances between the two vectors. Experiments on the scale coefficient show that the performance of this method can reach the best according to a ratio of 9:1 (cosine similarity: euclidean distance), and the mIoU reaches 40.5%. Extensive experiments are carried out on the plant disease dataset PDID-5i. From the experimental results, it can be seen that mIoU of our proposed network model is improved by 1.7% compared with that of the original PANet, which proves that our model has excellent performance in segmenting plant leaf disease infected regions. It provides the possibility for the prevention and control of early plant infection diseases, and can reduce economic losses and increase yield.
For details, please refer to page 13 line 432-446.
---------------------end---------------------
We appreciate very much for your kind suggestion and make all revisions in the updated manuscript. Thank you!

Reviewer 3 Report
Dear authors and editor,
the manuscript “Multi-scale and Multi-match for Few-Shot Plant Disease Image Semantic Segmentation” aims to identify plant diseases by a few-shot semantic segmentation model. Few-shot semantic segmentation is based on a few support images but does not obtain optimal results on plant disease segmentation. This paper attempts to implement a new methodology based on a multi-scale and multi-prototypes match to improve the model performance. The experiment aims to identify ten diseases. The results shown in the manuscript are encouraging and may help develop new tools for the farmers. I have some comments about the writing:
- The abstract should report the main results achieved in the study
- A discussion of the results is encouraged, e.g., pros and cons, lessons learned and practical implications of the work
- Check the order of the citations: numbering must be consecutive
- Revise English
Author Response
Manuscript ID: agronomy-1964305
Manuscript Title: Multi-scale and Multi-match for Few-Shot Plant Disease Image Semantic Segmentation
We submitted the revised manuscript; the revisions were carried out based on the editor and each reviewer’s comments. We thank the editor and all reviewers for their valuable comments and suggestions. The following summarizes our point-by-point responses to comments and suggestions.
---------------------------------------------
<< Author's Reply to the Review Report (Reviewer 3)>>
- The abstract should report the main results achieved in the study
Thank you very much for your valuable suggestions, and we supplement the main results of the study in the abstract.
For details, please refer to page 1 line 29.
- A discussion of the results is encouraged, e.g., pros and cons, lessons learned and practical implications of the work
Thanks for your valuable advice, we have re-written a discussion, adding some research findings, as well as the impact of work on agriculture and some of the inadequacies of our work.
We propose a novel multi-scale and multi-prototype matching few-shot segmentation network. First, the MPM structure in the network obtains prototype and query feature at two different scales through multi-scale feature fusion, which can enrich the relationship between leaf disease prototypes and query feature, thereby improving the accuracy of the model to identify disease areas. Furthermore, this paper proposes a new method for establishing the relationship between prototype and query feature maps, which takes into account the difference computation of angles and distances between the two vectors. Experiments on the scale coefficient show that the performance of this method can reach the best according to a ratio of 9:1 (cosine similarity: euclidean distance), and the mIoU reaches 40.5%. Extensive experiments are carried out on the plant disease dataset PDID-5i. From the experimental results, it can be seen that mIoU of our proposed network model is improved by 1.7% compared with that of the original PANet, which proves that our model has excellent performance in segmenting plant leaf disease infected regions. It provides the possibility for the prevention and control of early plant infection diseases, and can reduce economic losses and increase yield.
For details, please refer to page 13 line 432-446.
- Check the order of the citations: numbering must be consecutive
Thank you very much for pointing out the order of the citations, we have revised the numbering problem in time.
- Revise English
Thank you very much for your valuable suggestions on this article, there are some English grammar problems in this article, we have rechecked it and corrected the obvious mistakes.
---------------------end---------------------
We appreciate very much for your kind suggestion and make all revisions in the updated manuscript. Thank you!

Round 2
Reviewer 1 Report
Check "Ground Tures" in Figure 5.